# When Does Being Watched Change Pro-Environmental Behaviors in the Laboratory?

Cameron Brick [1,2,*] and David K. Sherman [2]

1   Department of Psychology, University of Amsterdam, Nieuwe Achtergracht 129 B,
    1018 WT Amsterdam, The Netherlands
2   Psychological & Brain Sciences, University of California, Santa Barbara, CA 93106, USA;
    david.sherman@psych.ucsb.edu
*   Correspondence: brickc@gmail.com

**Abstract:** Pro-environmental behaviors such as conserving water, reducing flights, or purchasing energy-efficient appliances are subject to social pressures. The influence of norms on behavior is widely studied, but it is less clear which social identities (e.g., political ideology; being an environmentalist) and contextual factors lead individuals to pursue or avoid pro-environmental behaviors. The visibility of behaviors—whether an action can be observed by others—has attracted wide research attention in psychology, business, and economics for theoretical and practical reasons. This paper includes three experiments on visibility, total $N = 735$ (U.S. university students). There were no effects of visibility on "green" purchases, donation to a conservation organization, or willingness to sign up for a water-reducing student meal plan; these null effects are consistent with a recent Registered Report. Additional predictors are also modeled, such as Openness and the need for status. It remains likely that being observed by certain audiences will affect certain pro-environmental behaviors in certain contexts. The discussion centers on methodological and conceptual issues contributing to null effects and to how future research can usefully explore individual difference moderators, type of audience, and types of pro-environmental behavior that influence when visibility might change conservation behaviors.

**Keywords:** social desirability; observability; pro-environmental behavior; identity; environmentalism; norms



## 1. Introduction

Perceived norms and social identities change individuals' perceptions, feelings, and behaviors, even in private [1–3], and these social pressures are routinely underestimated [4]. People may increase or decrease pro-environmental behaviors to bolster a valued identity or avoid signaling an unwanted association [5], with or without conscious awareness.

Pro-environmental behaviors exist in a changing spectrum of social acceptability from widely expected (e.g., not littering) to widely despised (e.g., veganism) [6]. Each behavior may have unique links to relevant social groups and identities, as well as contexts of particular importance. Nonetheless, researchers have attempted to use this diversity to uncover the social pressures on pro-environmental action. There is conflicting evidence about whether environmentalists are seen as generally cooperative and prosocial. One paper found that Canadian undergraduates viewed environmentalists negatively, e.g., as militant and eccentric, and that people avoid interacting with them because of these stereotypes [7]. However, a recent social dilemma experiment showed that a young sample of Austrians viewed environmentalists as cooperative and preferred them as partners in a non-environmental task [8].

Individuals strike a balance between assimilating to groups, which provides belonging and meaning, and differentiating themselves as individuals, which provides autonomy and self-clarity [1]. Whether from subgroups or society as a whole, social norms appear

to be uniquely powerful for shaping pro-environmental behavior [9]. Different groups have different norms that can either reduce or increase pro-environmental behaviors in those groups. The three current studies were conducted on a large University of California campus, where many students consider themselves "green", as shown by environmentalist identity being above the midpoint in all three studies. Anecdotally, the campus also has active environmentalist groups and is in a region that often discusses environmental issues such as drought, wildfires, and climate change.

Due to the importance of social norms, particular focus has been applied to revealing which social identities are most important (e.g., political ideology) [10,11] and what features of a situation would increase or decrease behaviors. If such factors were shown to be reliably associated with changes in pro-environmental behavior, this could inform interventions such as messaging and choice architecture. One feature that is both theoretically and practically important for testing social effects is how visible a pro-environmental behavior is to other people.

### 1.1. Social Visibility

Behaviors that can be observed by others are the behaviors most likely to be changed by social cues and pressures. Visibility effects are most likely for the behaviors that individuals view as environmentally relevant, rather than the behaviors with the greatest environmental impact. Even posters of watching eyes reduce anti-social behavior according to a recent meta-analysis [12]. In the environmental sphere, consumers pay more for clearly identifiable hybrid cars such as the Toyota Prius® compared to conventional-looking hybrid cars in politically left areas of the U.S. [13], and the density of nearby home solar panels increases local installations in California [14]. In contrast, how much one's neighbors conserve water in the home or how far they drive each year are harder to observe. As a result, those behaviors may be less influenced by norms, including perceptions of what people are actually doing and what people think others should be doing, although norms still influence such behaviors.

Early studies on public vs. private pro-environmental behaviors relied on text-based social primes and hypothetical intentions [15]. Later work supported the claim that individuals strategically pursue and avoid pro-environmental associations [16,17], but many papers were based on self-reported behavior and did not include experiments [18]. Field studies and objective measures of behavior are rare, but see [17]. As pro-environmental behaviors differ in their acceptability and members of different social groups will be incentivized to either pursue or avoid such behaviors, merely being watched may not produce consistent effects on a certain behavior. One key test is whether individuals who see themselves and want to be seen as environmentalists perform such behaviors more when watched, and individuals who do not want to be seen as that group would perform such behaviors less when watched [18].

This visibility question is unresolved and has theoretical and applied importance. Although early work in the U.S. pointed to political ideology as a key identity for pro-environmental behavior [16], recent studies suggest environmentalist identity may be even more closely related [10,18], which makes sense given the conceptual overlap between that group and pro-environmental behaviors. Additionally, a range of other identities have consistent but weaker relationships to self-reported pro-environmental behavior, such as being female [19]. Now that the identity space is better understood, there is a need to study how social contexts determine pro-environmental behavior.

Social identities can lead to pro-environmental behaviors through identity consistency, as a guideline for desirable actions [3], and through identity signaling, to boost one's reputation or improve social ties [20]. The visibility of behaviors is a uniquely powerful tool for testing the strength of these effects across different behaviors and contexts, because of the strength of motives to signal positive attributes to others [5,21]. Identity signaling and observability of pro-environmental behaviors were first investigated in three cross-sectional studies using multi-level modeling to account for individual ratings of behavior

frequency, visibility, difficulty, and conservation effectiveness in online samples [18]. Across 21 pro-environmental behaviors, this research found that non-environmentalists avoided visible behaviors (e.g., reusable grocery bags) (in 2 of 3 studies) and environmentalists engaged more in visible behaviors compared to less visible behaviors (e.g., home water use) (in 1 of 3 studies).

Another recent paper supports this account. In 1000 U.S. residents roughly representative to national demographics, environmentalist identity was a strong, unique predictor of a composite of pro-environmental behavior and its influence depended on whether the behavior was public or private. In a regression with other known predictors such as environmental attitudes and demographics, environmentalist identity was one of the strongest predictors of public environmental behaviors (partial $\eta^2$ = 0.15) but not private environmental behavior (partial $\eta^2$ = 0.01) [22].

The current paper tests the visibility effect in experiments designed to include more objective indices of pro-environmental behavior. Crucially, we expected visibility effects where social norms support green behaviors, like in the university student population tested here. University students are less diverse in age, education, and political ideology than the MTurk workers from studies like in [20], and the green norms may be stronger in university samples. Therefore, each study will also measure environmentalist identity.

**Hypothesis 1 (H1).** *Being watched will lead to more pro-environmental behavior (when pro-environmental behaviors are normative).*

A recent study provides some of the strongest evidence to date because it was a well-powered Registered Report examining an objective pro-environmental behavior in the laboratory in a pro-environmental population and context (*N* = 176) [23]. Participants completed a task that repeatedly pitted the convenience of reducing the study time against using energy to illuminate lights and therefore release unnecessary greenhouse gases (participants were informed of the environmental consequences). Participants were mostly Belgian university students who completed the task in a separated cubicle or next to another workstation where another student could observe their actions (and was informed about the meaning of the lights). The visibility condition had no effect on the frequency of pro-environmental behavior. The interaction between being watched and environmentalist identity could not be tested in the Registered Report because of low power: interactions require much larger samples than main effects, perhaps even 8–16 times more [24].

In the interest of better identifying the boundary conditions and moderators behind any effect of being watched, null effects are helpful such as in [23] that Registered Report. Emptying the file drawer of experiments [25] can inform the best studies to be replicated [26] and provide better meta-analytic estimates [27]. In that spirit, three experiments are shown below that tested for and did not find visibility effects, each with distinct manipulations of visibility and pro-environmental preferences or objective behaviors. There is no best-practices manipulation of visibility, so we used three different techniques and consider the overall pattern more informative than any individual study. In particular, these manipulations were designed for realism and similarity to plausible real-world events. Similarly, given the lack of consensus on taxonomies of pro-environmental behaviors, we used different behaviors in each study, updating the techniques over time based on the findings of each study.

Study 1 asked students to physically handle and then rate office products, some of which were eco-friendly (e.g., made from recycled materials). Visibility was manipulated by the presence or not of a research assistant in the room. In Study 2, participants were given the opportunity to donate to an environmentalist student group, and visibility was manipulated in the cover story by the donation either being published with their name or being completely private. Study 3 occurred during a severe drought and measured students' willingness to sign up for a water-saving meal plan, and visibility was manipulated with a proposal that included a large green sticker on their student identification card compared to no sticker. In each experiment, environmentalist identity was measured in a separate

prescreening to obscure the environmental focus of the studies. Of the three studies, only one found a main effect of environmentalist identity (Study 3) and none found a main effect of being watched.

### 1.2. Personality Predictors of Pro-Environmental Behavior

A secondary goal of these studies was to continue charting the associations of different pro-environmental behaviors with personality traits such as the Big Five [28] and the need for social status [15]. Studies 2 and 3 include Openness, Conscientiousness, Extraversion, Agreeableness, and Neuroticism. Most previous work highlights Openness as the personality trait most linked to pro-environmental attitudes and behavior [29,30], perhaps because abstract, flexible thinking is helpful to appreciate long-term and long-distance problems such as climate change. Additionally, the Openness facet Aesthetic Appreciation, which is characterized by enjoying aesthetic and sensory experiences such as listening to music or going to a museum, appears strongly linked to environmentalism [31]. Most of the related literature is based on self-reported behavior, so the current studies provide an opportunity to evaluate associations between personality and objective pro-environmental behavior.

### 1.3. Open Data, Code, and Materials

The below studies are from 2014–2015 and the lead author's dissertation. There were no public pre-registrations, power analyses, nor detailed analytic plans, but data collection was finished before analyses and care was taken to minimize branching analytic decisions and therefore false positive results in the current report. Alpha ($\alpha$) was set to 0.05 throughout and there were no corrections for multiple comparisons.

Separately, we ran a failed online replication of [15] using hypothetical product preferences, and it is omitted for space and clarity and to focus here on objective behaviors. A further visibility experiment was run in a collaboration at another university in 2015, but incomplete documentation prevented proper analysis and so it was also omitted. Otherwise, the current paper includes all unpublished experiments on visibility from the first author. Anonymized data, analysis code, stimuli, questionnaires, and further details are publicly available at the Open Science Framework (OSF) https://osf.io/hqnv2. Otherwise, the current paper includes all extant unpublished experiments on visibility from the lead author.

## 2. Study 1

Consumer behavior impacts environmental quality and is a key area of interest in environmental psychology [5,32,33]. Study 1 investigated preferences for sustainable, "green" office products relative to comparable, typical products from the campus bookstore. To boost plausibility and realism of the outcome measure, all the products were handled in person in the laboratory. In previous work, visibility of a behavior has sometimes been operationalized with watching eyes posters [12], text vignette priming [15], or reports of which behaviors are more visible than others [18]. However, real-world observability requires performing a behavior in the laboratory when another person is watching or not. Therefore, visibility was manipulated by participants giving their preferences either privately or in the presence of a student research assistant. The main outcome variable was the relative preference for "green" vs. typical products. As exploratory outcome measures, participants also chose a favorite product to receive as a gift (the chosen item was "green" or not), and then reported their satisfaction with a surprise "green" gift they received instead of their choice.

### 2.1. Methods

A pilot study described in the OSF materials was a laboratory study of 94 undergraduates that determined the selection of the experimental stimuli.

### 2.1.1. Participants

Study 1 included 226 undergraduates, 72.4% female, 27.5% male; 37.8% White/ Caucasian, 0.9% Black/African-American, 33.3% Asian/Asian-American, 25.3% Hispanic/ Latino, and 2.7% Other (0.2% no response); age $M$ ($SD$) = 19.3 (1.19), who completed prescreening at the beginning of a ten-week academic term. During the term, students participated for course credit in a laboratory session containing two ostensibly separate studies on a computer. 14 additional participants were excluded for not completing the study.

### 2.1.2. Procedure and Measures

**Visibility manipulation.** Participants saw a screen which read "Loading . . . " for 10 s, and then were confronted with a fabricated computer error (see OSF for all stimuli). This error both notified the research assistant of the randomized visibility condition and provided a plausible explanation for the presence of the research assistant in the room. In the public condition, the research assistant explained that they would remain in the room to monitor for other computer errors, and all the measures that followed were closely observed by the research assistant. In the private condition, the research assistant explained that the error was unlikely to happen again and left the room.

**Prescreening.** Environmentalist identity was measured in prescreening weeks prior to avoid suspicion in four items [18]: I see myself as pro-environment; I am pleased to be pro-environment; I feel strong ties with pro-environment people; I identify with other pro-environment people, all rated 1 (*disagree strongly*) to 7 (*agree strongly*); Cronbach's $\alpha$ = 0.88. The distribution had low normality, skew = $-0.92$ and kurtosis = 2.95, due to peaks at the lowest and middle values. Exploratory scales such as climate change belief and other identity measures were also included (see OSF).

**"Green" product preference (primary outcome).** Participants physically handled each bookstore product (e.g., pens; post-it notes; see Supplementary Figure S1 and OSF for all stimuli) and then answered two questions: "How much do you like this product?" rated 1 (*Dislike strongly*) to 7 (*Like strongly*), and "Would you like to own this product?" rated 1 (*Not at all*) to 7 (*Very much*). These two items correlated $r$s(223) = 0.63–0.87 across the three studies here, $p$s < 0.001, and were combined into a composite of product preference. A single value of relative preference for "green" products was computed by the mean of the product preference ratings for the "green" items minus the mean of the product preference ratings for the typical items.

**Filler task.** Participants completed a brief filler task of solving five easy items from the Remote Associates Task [34]. Participants were then told they had completed the first study and were shown a second consent form for an ostensibly separate task.

**Exploratory outcome: preference for "green" gift.** Participant success on the easy Remote Associates Test provided a plausible reason why participants were told that their performance was "very good" and they would receive the product they marked earlier as their favorite. All participants were then apologized to and told that the item they selected was not available, and at the end of the session they were given a small recycled paper notebook (see OSF for all stimuli). Participants responded to two questions, "How satisfied are you with your prize?" and "How pleased are you to own this prize? rated 1 (*not at all*), 2 (*a little*), 3 (*medium*), 4 (*a lot*), or 5 (*extremely*), which correlated $r$(223) = 0.83, $p$ < 0.001 and were combined into a composite of gift satisfaction.

### 2.1.3. Exploratory Measures

**Environmental attitudes**. The 14-item Connectedness with Nature Scale [35] taps affective as well as cognitive content with items such as: "I often feel a kinship with animals and plants" and "I have a deep understanding of how my actions affect the natural world", rated 1 (*Strongly disagree*) to 5 (*Strongly agree*), Cronbach's $\alpha$ = 0.82.

**Political ideology.** Party identification was measured and calculated as in the American National Election Studies [36]. If participants selected a political party affiliation of Democrat or Republican, they next indicated the strength of their party affiliation from

1 (*Not very strong Democrat/Republican*) to 7 (*Strong Democrat/Republican*). If participants first selected Independent or another category (e.g., Green Party), they rated their party preferences from 1 (*Strong Democrat*) to 7 (*Strong Republican*). All people's ratings were combined to yield a common rating of political liberalism from 1 (*Strong Republican*) to 7 (*Strong Democrat*).

**Suspicion, feedback, participant code, and other.** Participants answered two funnel suspicion prompts, related any technical problems, and entered their student identification number in order to link their data to prescreening. Other, exploratory measures were collected but are not discussed here: self-reported visibility and frequency of pro-environmental behaviors as part of a separate, multilevel replication; other demographics; and ratings of the research assistants (perceived environmentalism and attractiveness; see OSF for all materials).

### 2.2. Results

Environmentalist identity was moderate, $M$ (*SD*) = 3.54 (1.82). On its face, this result may not seem consistent with the argument that university samples have stronger green norms than other populations like MTurk, but this may obscure that participants are using different reference groups for what they consider an environmentalist. Participants slightly preferred "green" compared to typical products, "green" preference $M$ (*SD*) = 0.57 (0.99); 46.0% selected a "green" product to receive as a gift (two of six items were green; random would have been 33.3%), and satisfaction with the surprise "green" gift was moderate, $M$ (*SD*) = 3.45 (0.91). None of these outcome variables was correlated with environmentalist identity, $p$s ≥ 0.07.

The distribution of identity was non-normal (see Methods), so an exploratory analysis was conducted excluding the participants who answered the lowest value to all four questions or the middle value to all four questions, which was interpreted as them not taking the task seriously. The remaining sample of 135 participants were assumed to be providing more valid data. This exploratory identity measure predicted "green" product preference, $r(133)$ = 0.21, $p$ = 0.02, but not the two other outcomes, $p$s > 0.39 (see Table 1). All participants were retained for the main analysis below.

**Table 1.** $M$, *SD*, scale reliability, and zero-order correlations between environmentalist identity, environmental attitudes, political orientation, and "green" product preferences (Study 1).

| $r(224)$ | 1 | 2 | 3 | 4 | 5 | 6 |
|---|---|---|---|---|---|---|
| $M$ | 3.54 | 3.43 | 3.34 | 0.57 | 0.54 | 3.45 |
| (*SD*) | 1.82 | 0.61 | 1.28 | 0.99 | 0.50 | 0.91 |
| Cronbach's $\alpha$ | 0.84 | 0.82 | | | | |
| 1: Environmentalist identity (1–7) | | | | | | |
| 2: Environmental attitudes (1–5) | 0.21 ** | | | | | |
| 3: Political liberalism (1–7) | −0.06 | −0.15 | | | | |
| 4: "Green" product preference (1–5) | −0.02 | 0.06 | 0.03 | | | |
| 5: "Green" favorite item (0,1) | −0.08 | 0.03 | 0.16 | 0.41 *** | | |
| 6: Satisfaction with "green" gift (1–5) | 0.12 | 0.23 *** | −0.15 | 0.09 | -0.05 | |
| 7: Visible behavior condition (0,1) | 0.01 | −0.02 | 0.04 | −0.13 | −0.21 ** | 0.04 |

Note. * $p$ ≤ 0.05, ** $p$ ≤ 0.01, *** $p$ ≤ 0.001.

**Main effect (H1)**: Hypothesis 1 was that when pro-environmental behaviors are normative, being watched will lead to more pro-environmental behavior. In this university context, when behaviors are visible the assumed norm would be to act "green". The main outcome was green product preference, which was computed by comparing the mean preferences for "green" vs. non-"green" items (see above). Green preference was similar in both conditions (range −1.75 to 3.5), public $M$ (*SD*) = 0.44 (0.92) ($n$ = 114),

private $M$ ($SD$) = 0.70 (1.04) ($n$ = 112). An independent samples $t$-test failed to reject the null hypothesis; rating the products in private vs. public was unrelated to preferring "green" products, $t(184) = -0.48$, $p = 0.63$. The exploratory outcome of picking a favorite item that was "green" (yes or no) provided an alternate test that also contradicted H1, public $M$ = 43.9%, private $M$ = 64.3%. See Supplementary Figure S4 for an underpowered interaction of environmentalist identity and visibility condition on behavior.

*2.3. Discussion*

Study 1 examined the effect of being watched when evaluating and receiving campus bookstore products. It was concealed at every stage that the study was about environmentalism, from prescreening to recruitment to the laboratory procedure (e.g., most of the products were conventional) to reduce experimental demand that could influence the participant's perception of experimenter expectations. This concealment may have decreased the salience of the environmental decisions so much that the students may not have realized the behaviors were relevant to the environment. Environmentalists in the study did not prefer the "green" products, choose a favorite item that was "green," nor express more satisfaction with the "green" notebook gift than non-environmentalists. The original logic is based on behaviors that environmentalists would like to do more than non-environmentalists, and therefore the below studies used different outcome measures.

**3. Study 2**

Study 2 examined donation to an environmental cause. Donation is an objective pro-environmental behavior that can be completed in the laboratory [37,38]. In order to manipulate visibility and set up the donation without alerting participants to the environmental focus of the research, an elaborate cover story was generated about evaluating student groups on campus. The visibility manipulation was designed to give students a sense of their actions being visible to their peers that was more visceral than a priming vignette. Students were told that their donation would be either completely private or that donor names would be publicized online, and in the latter condition were shown an example screenshot of student names on the website (these names were fictitious). As an exploratory aim, personality was assessed to follow up on links between Openness and pro-environmental behavior.

*3.1. Methods*

In the laboratory, participants completed two personality questionnaires and then responded to a survey about student groups, after which they were surprised with ten raffle tickets for the chance to win $50 and had the opportunity to donate tickets to an environmental student organization. As a manipulation of visibility for the donation, participants were randomized to have their decision be completely private or publicized online.

3.1.1. Participants

159 undergraduates, 67.5% female, 32.5% male; 35.8% White/Caucasian, 35.2% Asian/ Asian-American, 3.8% Black/African-American, 23.9% Other (1.3% no response); 25.2% Hispanic or Latino; age $M$ ($SD$) = 18.8 (1.21), completed prescreening at the beginning of a ten-week academic term. Later that term the students participated for course credit in a laboratory session containing two ostensibly separate studies on a computer. The separation of prescreening and the two sections allowed the laboratory session to obscure that the research was about environmentalism and to improve the plausibility of the cover story.

3.1.2. Procedure

After the personality measures, participants were told they were beginning a different study and providing opinions about student groups in collaboration with the university (see Supplementary for the cover story). Students were told that a student group was

ostensibly selected at random for their evaluation, but all students saw a description of and completed some filler items rating the Environmental Affairs Board, a real and active group on campus.

**Environmental donation.** Participants received ten raffle tickets for a $50 prize and chose to donate 0–10 to the student group, with instructions: "Associated Students is sponsoring a raffle for $50 for survey respondents, and for participating today you earned ten raffle tickets. Congratulations! The winner will be announced after data collection is complete (likely Winter quarter). You have the opportunity to donate any or all of your raffle tickets to the organization you gave feedback about. Your group was: ENVIRON-MENTAL AFFAIRS BOARD". The winning ticket was selected randomly. The participant had donated the winning ticket to the organization, so to the considerable surprise of the student group we gave them a $50 Amazon.com gift card.

**Visibility manipulation.** Participants were randomized between groups. Private condition: "Your donation is anonymous. How many tickets would you like to donate to this group? Pick any number between 0–10. The research assistant cannot see your decision". Public condition: "If you donate one or more tickets, your name will be listed on the Associated Students webpage as a "valued supporter" of ENVIRONMENTAL AFFAIRS BOARD. Your name will also appear on a thank-you webpage of this group, and all students can see these webpages. This is how the thank-you page will appear: [Figure 1]. How many tickets would you like to donate? If you give one or more tickets to this group, you will be thanked in public for your support. Pick any number between 0–10. The research assistant cannot see your decision".

### 3.1.3. Measures

**Environmentalist identity and political ideology.** See Study 1.

**Personality.** A Big Five measure of personality was completed as an exploratory measure [28] to follow up on links between Openness and pro-environmental behavior [31].

**Need for social status.** Four items were used (see OSF for full items). The scale had poor reliability, Cronbach's $\alpha$ = 0.54, indicating some uncertainty as to what construct or constructs were measured by the four items, and so the most face-valid item, "I want people to know that I am an important person of high status," was used in place of a composite in the analyses.

**Environmental attitudes.** Fifteen items of the New Ecological Paradigm [39] yielded adequate reliability and were combined into a composite, Cronbach's $\alpha$ = 0.75.

**Demographics, quality check, and attitudes.** Finally, participants reported their age, gender, parents' household income, race, ethnicity, and student number to connect their results to prescreening. Participants were also asked if they had technical problems with the survey, responded to two questions of a funnel suspicion prompt to indicate their recognition of the study deception, and were invited to give comments for our team before debriefing. Another, exploratory measure of attitudes about environmentalists was collected but is not discussed here (see OSF).

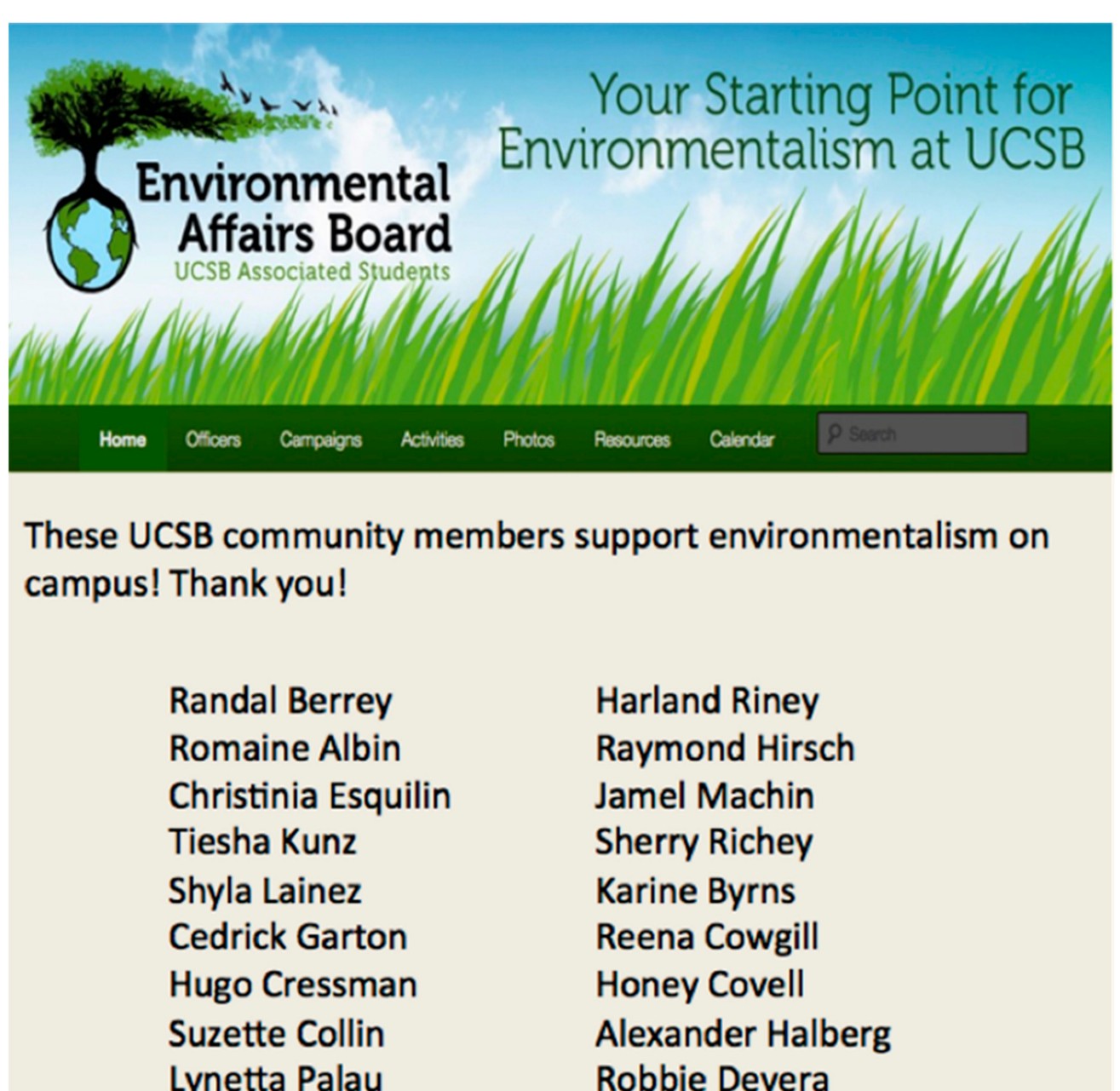

**Figure 1.** Visibility was manipulated in Study 2 by some participants being randomized to a condition where any donation would mean their name would be published online (the displayed names were fictitious).

*3.2. Results*

Environmentalist identity was moderate, *M* (*SD*) = 3.85 (1.20). Identity was again not optimally distributed for linear analyses due to being peaked at the scale midpoint, skew = −0.51, kurtosis = 2.98. The length of the prescreening procedure weeks before (about 300 questions done online) may have reduced participant attention and carefulness, and therefore increased noise such as repeatedly selecting the midpoint value, which could have resulted in an exaggerated peak at the midpoint. Means, variance, and zero-order correlations are shown in Table 2.

**Table 2.** *M*, *SD*, scale reliability, and zero-order correlations between key variables (Study 2).

| *r*(157) | 1 | 2 | 3 | 4 | 5 | 6 |
|---|---|---|---|---|---|---|
| *M* | 3.85 | 3.63 | 3.60 | 4.71 | 3.67 | 3.61 |
| (*SD*) | 1.20 | 0.47 | 0.53 | 0.93 | 1.36 | 3.90 |
| Cronbach's α | 0.91 | 0.75 | 0.70 | 0.90 | | |
| 1: Environmentalist identity (1–7) | | | | | | |
| 2: Environmental attitudes (1–5) | 0.41 *** | | | | | |
| 3: Openness (1–5) | 0.38 *** | 0.17 * | | | | |
| 4: Attitudes towards env'ists (1–7) | 0.60 *** | 0.40 *** | 0.37 *** | | | |
| 5: Political liberalism (1–7) | 0.13 | −0.09 | −0.05 | −0.05 | | |
| 6: Environmental donation (0,1) [a] | −0.02 | −0.14 | −0.05 | 0.02 | −0.00 | |
| 7: Visible behavior condition (0,1) [a] | −0.05 | −0.18 * | 0.07 | −0.02 | −0.10 | −0.06 |

Note. * $p \leq 0.05$, *** $p \leq 0.001$; [a] Dichotomous; row values are point-biserial correlations.

**Environmental behavior.** Donation to the environmental organization was moderate, $M$ (*SD*) = 3.61 (3.90) of ten raffle tickets. The distribution was U-shaped and appeared bimodal at zero and the scale midpoint (5); skew = 0.73 and kurtosis = 1.90. Many participants likely chose zero out of self-interest, and a donation of five may have been more common than four or six due to splitting the difference between self-interest and pro-social goals. Due to these two modes, using donation as continuous would violate assumptions in the general linear model. Donation was dichotomized based on the shape of the modes with 1–5 recoded to 0 ($n = 117$), and 6–10 recoded to 1 ($n = 42$) prior to testing the hypotheses (see below for a non-parametric test without dichotomization). To account for the possibility that donation scores of 5 were qualitatively different than 0–4, an exploratory analysis excluded the "5" values and tested 0–4 against 6–10 and it revealed similar results. All analyses were rerun comparing raw vs. dichotomized donation amounts and the results were similar. See Supplementary Figure S5 for an underpowered interaction of environmentalist identity and visibility condition on behavior.

**Main effect of visibility (H1):** Donation (dichotomized) was similar in both visibility conditions, public = 23.8% ($n = 80$), private = 29.1% ($n = 79$). A chi-squared test failed to reject the null hypothesis; dichotomized donation was unrelated to the private vs. public condition, $\chi^2(1) = 0.34$, $p = 0.56$. H1 was also evaluated in a logistic regression where dichotomized donation amount was predicted from environmentalist identity, environmental attitudes, attitudes about environmentalists, political orientation, and visibility condition. The sole finding was a main effect of environmental attitudes, $OR$ (151) = −1.04, $SE = 0.48$, $z = −2.18$, $p = 0.03$, contrary to expectations, such that individuals higher in environmental attitudes donated less often; all other non-intercept $p$s ≥ 0.31. As a further robustness check, a non-parametric Mann–Whitney U test was performed on the full distribution of donation with condition as a predictor, and there was no effect of condition, Wilcoxon $W = 3343$, $p = 0.52$.

**Exploratory personality analyses.** Contrary to expectations, the only personality trait from the Big Five and the single status item to correlate with raw donation amount was Agreeableness, $r_{pb}(157) = 0.18$, $p = 0.02$, indicating that individuals who were high in Agreeableness were more likely to donate to the environmental organization (point-biserial correlation). In a logistic regression with the six personality predictors, Agreeableness was the only main effect, $OR$ (157) = 1.64, $SE = 0.37$, $z = 2.21$, $p = 0.03$.

### 3.3. Discussion

Study 2 created a successful cover story (based on responses to the suspicion prompt), observed a real-world environmental behavior, and leveraged prescreening methods to conceal that the study was specifically about environmentalism. However, none of the main predictors correlated with environmental donation. The key individual difference predictor

of environmentalist identity had a potentially non-normal distribution, as in Study 1. A plausible explanation for the sharp peak at the midpoint of the scale is satisficing (low task engagement causing low measurement validity). If participants were not engaged, this would reduce the ability to test the hypotheses. The outcome variable was also non-normal, appeared bimodal, and was dichotomized as a result. However, this dichotomization obscured the full range of donation choices, and therefore also reduces the power to test the hypotheses. Based on the lack of correlations of the outcome variable with environmentalist identity and attitudes, the donation may not have been seen as environmental. It is possible that the decision of how much to donate activated other, more dominant schemas for students than environmentalism. There was also a concern about the unexpected correlation between attitudes and experimental condition, $r(157) = -0.18$; this could either be spurious or indicate a failure of random assignment.

Exploratory analyses revealed that highly agreeable individuals—those who seek interpersonal harmony and want to please others—donated more to the environmental group. Since the behavior may not have been seen as environmental, it follows that the behavior was interpreted broadly as a pro-social opportunity and the agreeable individuals most interested in being prosocial donated more. This is also consistent with a previous finding. The absence of a relationship between Openness and donation further suggests the behavior was not seen as environmental, since that link is well-established [29,31]. In sum, Study 2 was not a definitive test of H1, likely due to issues of operationalization and experimental procedure. As a result, Study 3 used a new visibility paradigm and new pro-environmental behavior outcome.

## 4. Study 3

Study 3 presents a new design intended to improve key variable distributions. To test whether prescreening is causing the non-normal distributions in identity, Study 3 also includes a second measurement of environmentalist identity during the laboratory session. As environmentalist identity did not predict green behavior in Studies 1 and 2, it is unclear whether the outcome behaviors were seen as "green" by the students. Conceptually, a behavior in the laboratory may seem more "green" to the extent it is seen as addressing a serious environmental problem. Based on this logic, Study 3 uses a new measure of whether students are willing to change their meal plan to reduce water use during a state-wide drought. As this behavior is more explicitly justified and linked to environmental outcomes, it was expected to relate more closely to environmentalist identity than the behaviors in Studies 1 and 2. Additionally, it was unclear in Study 2 how much the online publication of names felt public to the students. Therefore, the public manipulation of visibility was changed to a large sticker on the student identification card, and students were shown this sticker graphic, with the goal of increasing the sense of public visibility in that condition.

Study 3 also continued the investigation of personality. The secondary predictions were that Openness would be the Big Five trait most associated with pro-environmental behavior, and based on Study 2 Agreeableness would also predict the environmental behavior.

During data collection in February and March 2015, California was in an extended, extreme drought [40]. Undergraduate students at this campus were likely aware of the drought due to extensive public outreach from the local water board, signs and communications from the university, and an executive order from the state governor to reduce residential water usage. Study 3 uses the context of the drought to provide an opportunity for a realistic and timely environmental behavior: selecting a university meal plan for the following year that uses less water ("The Green Plan").

### *4.1. Materials and Methods*

4.1.1. Participants

The final sample was 350 undergraduates, 68.3% female, 31.4% male, 0.3% no response; 35.7% White/Caucasian, 30.6% Other, 29.7% Asian/Asian-American, and 3.4% Black/African-American (0.6% no response); 32.3% Hispanic or Latino (0.3% no response); age *M* (*SD*) = 18.9 (1.21). Additionally, one participant was excluded for incomplete data, and 10 were excluded for not providing a valid student identification number to allow the merge with prescreening data. All exclusions occurred prior to hypothesis testing.

4.1.2. Procedure and Measures

Participants completed prescreening at the beginning of a ten-week academic term. Later, participants ostensibly completed two separate studies in a single laboratory session as in Study 2. In the laboratory, participants filled out two personality questionnaires and then were introduced to a survey supposedly from the dining hall organization of their school. They heard about a proposed meal plan and were randomized to conditions where the sustainable meal plan add-on was a private or a public choice.

**Environmentalist identity (prescreening).** To improve simplicity and face validity, the wording from Study 1 was changed to remove "pro-" from the items, which then subsequently read: "I see myself as an environmentalist"; "I am pleased to be an environmentalist"; "I feel strong ties environmentalists"; and "I identify with environmentalists", each rated from 1 (*Disagree strongly*) to 7 (*Agree strongly*). The four items were reliable, Cronbach's $\alpha$ = 0.92, and combined into a composite.

**Personality.** In the laboratory, the first measure was the 44-item Big Five Inventory (see Study 2) and it served to justify the ostensibly separate first study in this session as well as provide exploratory tests of association with environmental behavior. The primary prediction was that Openness would be the Big Five trait most associated with pro-environmental behavior [31]. A weak, secondary prediction based on Study 2 was that Agreeableness would be the personality trait most associated with the environmental behavior.

**Need for social status.** See Study 2 for the four items. The scale had poor reliability, Cronbach's $\alpha$ = 0.58, indicating some uncertainty or unintentional breadth, and so the most face-valid item, "I want people to know that I am an important person of high status," was used in place of a composite in the analyses. After this measure, the participants were led to believe they began a separate study (see Supplementary for full text).

**Drought report.** Next, participants were given true data showing the extreme drought conditions in California to further connect the Green Plan [40]. "Next, we would like to explain a change that is being considered by Dining Services to reduce water use, and get your opinion of a new meal plan option. This CA drought report was just released, and shows intense water scarcity all over the state". (Supplementary Figure S2).

**"Green" meal plan.** Participants were then introduced to a sustainable meal plan add-on. "Please carefully read the information below. We will ask later in the survey about the details below to make sure we're getting useful feedback. [School name] is experiencing water shortages and increased prices due to the ongoing California drought. The drought is expected to last several more years, and in response, Dining Services is considering adding a new Green Plan® option as an add-on for each of their meal plans. [School name] is committed to reducing water use, and this plan is designed to increase student awareness and engagement with conservation. The Green Plan® would support [School name] to source meat, fruits, and vegetables from more water-sustainable farms and companies. This plan will reduce indirect water use by the dining halls by 17% as part of [School name]'s broader sustainability initiative to reduce, re-use, and recycle. The more sustainable food would be served campus-wide and plan holders would eat from the same food as other students".

**Visibility manipulation.** Participants were randomized to groups. In the private condition, participants read: "The Green Plan® would be a completely private choice. No

indication would appear on your Access card, or to the dining hall employees when you check in, or to other students, or even on your yearly bill. If your parents contribute to the cost of your meal plan, they would see a line item about costs related to the drought, and the word "Green" would not appear. Do you understand it would be a completely private choice? [Participants selected *Yes* or *No* as a manipulation check; all participants marked *Yes*].

In the public condition, participants read: "If you select the Green Plan®, a large green sticker will be added to your Access card and it will be required to use your meal plan (Supplementary Figure S3). Do you understand this choice can be seen by other people? [*Yes* or *No* served as a manipulation check; all participants marked *Yes*].

**Environmental behavior.** "The Green Plan® will increase the cost of existing meal plans by \$30 per year. Imagine that you are buying a meal plan from [School name] for next year. You can select any of the meal packages. Would you personally add the Green Plan®, based on all the details you saw above? I would choose the Green Plan® next year," rated 1 (*Strongly disagree*) to 7 (*Strongly agree*). In both conditions, participants were asked: "What are your thoughts on the the Green Plan®? Please give feedback below in several sentences about why you like or dislike the proposal. Thank you". (In the first sentence of the original materials, "the" was inadvertently doubled.) This feedback was used to get a qualitative sense of student reactions. The comments were sincere and impassioned, both advocating for the drought and arguing against increased costs for students, reflecting that the cover story and manipulation appeared successful.

**Environmentalist identity (laboratory).** To measure the test-retest reliability of the central identity measure, it was re-administered at this point during the laboratory session. The length of time between pretesting and the laboratory study ranged from 2–7 weeks, $M$ ($SD$) = 4.0 weeks (13.5 h). The four items were reliable, Cronbach's $\alpha$ = 0.93, and combined into a composite. Unfortunately, due to a technical error this measure was only administered to participants in the public visibility condition ($n = 175$). However, reliability between the identity scales is shown below for the public condition.

**Attention check.** Three questions served as attention checks: "What is the Green Plan® designed to do?", with options: *Save electricity*; *Reduce carbon emissions*; *Save water*; *Increase recycling* and 98.0% answered "save water" correctly; "Can other people see if you have the Green Plan®? and 96.3% answered correctly based on their condition; and "How could other people find out if you had the Green Plan®?" with options: *Special green tray*; *Sticker on Access card*; *Green Plan® t-shirt*; *Separate tables in the dining hall*" and 97.1% answered correctly (public condition only). Overall, participants understood and remembered the cover story. The below analyses did not differ substantially if the failures were excluded, so they are retained.

**Demographics and quality check.** Finally, participants reported their political orientation (see Study 1), age, gender, parents' household income, race, ethnicity, and student number to connect their results to prescreening. Participants were also asked if they had technical problems with the survey, responded to two questions of a funnel suspicion prompt to indicate their recognition of the study deception (these responses were not formally coded; anecdotally the deception appeared successful), and then the participants were invited to give any other comments before debriefing.

*4.2. Results*

Environmentalist identity was moderate, $M$ ($SD$) = 4.03 (1.15). Identity was peaked at the scale midpoint, skew = −0.36, kurtosis = 3.12. The same scale was re-administered during the laboratory session $M$ ($SD$) = 40.0 (10.1) days later, and was also peaked at the midpoint, $M$ ($SD$) = 4.19 (1.25), skew = −0.30, kurtosis = 3.38. Typically, kurtosis values above three reflect severe violations of normality. However, inspection of these distributions suggest they are otherwise normally shaped, unlike the bimodal donation variable in Study 2. Therefore, they are included in the linear model analyses below, but the

results should be interpreted with caution. The two scales were moderately related, $r(147)$ = 0.57, $p < 0.001$, indicating modest test-retest reliability in this measure across 1–2 months.

**Environmental behavior.** Composite preference for "The Green Plan" was moderate, $M$ ($SD$) = 4.28 (1.83). The distribution was acceptably normal, skew = −0.33, kurtosis = 1.96. Means, zero-order correlations, and scale reliabilities are shown in Table 3. See Supplementary Figure S6 for an underpowered interaction of environmentalist identity and visibility condition on behavior.

**Table 3.** $M$, $SD$, scale reliability, and zero-order correlations between environmentalist identity measures, agreeableness, need for status, political orientation, selection of the environmental meal plan, and visibility condition (Study 3).

| $r(348)$ | 1 | 2 | 3 | 4 | 5 | 6 |
|---|---|---|---|---|---|---|
| $M$ | 4.03 | 4.19 | 3.68 | 2.30 | 3.55 | 4.28 |
| ($SD$) | 1.15 | 1.25 | 0.55 | 1.15 | 1.44 | 1.83 |
| Cronbach's $\alpha$ | 0.92 | 0.93 | 0.75 | | | |
| 1: Env. identity (prescreening, 1–7) | | | | | | |
| 2: Env. identity (laboratory, 1–7) [a] | 0.57 *** | | | | | |
| 3: Openness (1–5) | 0.20 *** | 0.25 *** | | | | |
| 4: Need for status (1–5) [b] | 0.00 | 0.05 | −0.06 | | | |
| 5: Political liberalism (1–7) | −0.22 *** | −0.08 | 0.07 | −0.03 | | |
| 6: Preference for "The Green Plan" (1–7) | 0.28 *** | 0.32 *** | 0.16 ** | −0.02 | −0.13 ** | |
| 7: Visible behavior condition (0,1) [a] | −0.07 | *n/a* | −0.03 | 0.04 | −0.06 | 0.03 |

Note. ** $p \leq 0.01$, *** $p \leq 0.001$. [a] Row correlations only reflect the public visibility condition ($n$ = 175). [b] Single item.

Preference for the green plan was similar in both visibility conditions (agreement ranged from 1 to 7), public $M$ ($SD$) = 4.34 (1.79) ($n$ = 175), private $M$ ($SD$) = 4.23 (1.86) ($n$ = 175). In a regression without covariates, there was a main effect of identity, $\beta$ = 0.88 ($SE$ = 0.32), $t(289)$ = 2.78, $p$ = 0.006, but not of visibility, $\beta$ = 0.10 ($SE$ = 0.21), $t(289)$ = 0.51, $p$ = 0.61.

**Exploratory regressions with covariates.** Additional regressions were run on preference for the Green Plan with environmentalist identity (pre-screening), visibility condition, and each of the Big Five and also need for social status with continuous measures standardized prior to analysis. The only main effect on preference for the green plan was environmentalist identity, $\beta$ = 0.60, $SE$ = 0.11, $t(284)$ = 5.72, $p < 0.001$, such that environmentalists agreed more with the green plan; all other $p$s $\geq$ 0.16. Visibility condition did not predict green plan preference either in that regression or as a zero-order correlation, $r(348)$ = 0.03, $p$ = 0.58.

**Exploratory personality analyses.** Two personality traits from the Big Five correlated with Green Plan preference: Openness, $r(348)$ = 0.16, $p$ = 0.003, and Agreeableness, $r(348)$ = 0.11, $p$ = 0.05. Agreeableness and Openness were positively related, $r(348)$ = 0.12, $p$ = 0.03. An exploratory linear regression with the Big Five traits and social status predicting green behavior showed only one unique main effect, for Openness, $\beta$ = 0.48, SE = 0.18, $t(343)$ = 2.65, $p$ = 0.009. Openness having the strongest relationship with pro-environmental behavior is consistent with previous research [29,31] and lends support to the idea that Study 3 measured a behavior that seemed pro-environmental to participants.

### 4.3. Discussion

Preferring a water-saving meal plan was a more valid pro-environmental behavior than in Studies 1 and 2 based on open-ended student comments and answers to the suspicion prompt, as well as the convergent validity results. Environmentalists were more

likely to prefer the "green" plan than non-environmentalists, and the personality trait most associated with preferring the "green" plan was Openness as expected.

However, the visibility manipulation of the "green plan" sticker may have been weaker than intended. A first possibility is that the cover story may not have seemed believable. The self-report and suspicion prompt answers argue against this interpretation. Second, participants may have believed the story but figured the action would not be seen by their key peers; this is a plausible interpretation from the study materials, and therefore deserves strong consideration. Future studies could consider and assess the perception of visibility specifically to persons or groups to whom the participant is concerned about signaling positive qualities. Third, the manipulation may have been believable and seen as real but not psychologically important: that is, perhaps in this population signaling environmental support is not surprising or meaningful. A second look at the student body suggests this interpretation is also plausible. This campus has a national reputation for sustainability, and multiple student groups work directly on conservation and environmental issues. The university administration is also openly committed to sustainability and posts signs around campus such as ones that advertise water savings from irrigating with recycled water. Therefore, signaling environmentalism may be normative and it is possible that even non-environmentalists at this school are not concerned about the social risk of displaying pro-environmental behaviors [5].

## 5. General Discussion

Three studies examined novel paradigms of public vs. private pro-environmental behaviors. Previous findings from different research groups using different measures, visibility operationalizations, and different samples found evidence that observability affects pro-environmental behaviors and most of all for the strongly identified [18,22]. Here, none of the studies found an effect of visibility on "green" preferences or objective pro-environmental behaviors. Below, we suggest explanations for these findings and propose empirical targets for this research area.

**Construct validity.** One possibility is that the unvalidated stimuli, manipulations, and behavior measures did not tap the intended constructs as intended [41]. We encourage future researchers to validate their stimuli, manipulations, and measures prior to hypothesis testing [42]. Study 1 used a novel product evaluation task with high ecological validity, but the environmental behavior of preferring "green" products might not have been interpreted as environmental by the participants based on the lack of a relationship between that behavior and environmentalism. Study 2 used an objective donation measure, which is a well-known laboratory operationalization of behavior, but since environmentalists did not engage in it more than non-environmentalists regardless of visibility condition, donation may have also been interpreted as less relevant to the environment than intended. Study 3 succeeded in measuring a behavior that was seen as environmental by participants—the adoption of a water-saving meal plan—but the manipulation of visibility may have been ineffective.

Additionally, none of the studies found the predicted interaction between environmentalist identity and visibility on behavior. A likely explanation is a lack of construct validity: the visibility manipulation was not meaningful for participants or the outcome measures were not linked to environmental concerns. Another possibility is that the interaction tests were underpowered [24]. Additionally, the vignettes might have been too subtle. As the prescreening separated the environmentalist identity measure in time, and there was no clear indication the studies were about environmentalism, participants may not have seen the tasks as relevant to conservation or environmental protection.

**Individual differences.** Studies 2 and 3 assessed personality traits. Since the donation behavior in Study 2 may not have been seen as environmental, it follows that the most agreeable participants donated more; the behavior was perhaps interpreted as an opportunity to be pro-social rather pro-environmental. In Study 3, more Openness predicted more green behavior, consistent with earlier studies on a range of pro-environmental behav-

iors [29,30]; this suggests that preference for the Green Plan was viewed as environmentally relevant. Across the three studies, environmentalist identity had non-normal distributions perhaps due to anchoring and satisficing, making it more difficult to detect relationships with behavior. Our recommendation would be to reduce noise and bias in this measure, for example by not assessing it during a massive online prescreening. Additionally, these studies were all on U.S. undergraduates, which has severe limitations on the generalizability to other countries and populations [43,44]. Each population has different norms and even a different strength of relationships between cognitive variables and behaviors [45]. Future work could prioritize diverse samples within and between underrepresented countries, particularly because there are individual differences in being motivated to signal certain qualities [46].

**Audience type.** There could be no influence of being observed on pro-environmental behavior, but this seems unlikely given previous findings, e.g., [47]. There is an ongoing need to reconcile the literature on identity signaling and on pro-environmental behaviors, e.g., [16,18] with the null effects of the current studies and others, e.g., [23]. A promising direction is to further measure and manipulate what audience is viewing the behaviors and how often. Therefore, evidence suggests that visibility of behaviors to strangers may not be particularly motivating [23], particularly in one-shot social interactions. This is consistent with a more abstract view of who people want to signal to, and why; signaling seems likely to be most important for repeated interaction partners. This could explain why visibility appears important in cross-sectional designs that measure existing social relationships at home and at work, e.g., [18]. It would be valuable for future experiments to find ways to create behaviors that can vary in visibility to these meaningful audiences.

**Pro-environmental behavior type.** The lack of clarity from these studies and others on observability of behaviors [23] highlights the need for new frameworks for classifying and validating types of pro-environmental behavior. A recent review helps survey this area [38], and there is evidence that self-reported pro-environmental behaviors are only modestly related to objectively measured behaviors [48], which is a major concern for studies that infer behavior from intentions or self-report. There is also a need for studies that assess multiple behaviors and identities to inform how much diverse behaviors are caused by environmentalism.

In the meantime, it remains an open question when being watched leads to more or fewer pro-environmental behaviors. We hope the suggestions above will help resolve these questions and spur conservation behaviors. After all, someone is watching us.

**Supplementary Materials:** The following are available online at https://www.mdpi.com/2071-1050/13/5/2766/s1. Figure S1: Recycled paper clips were a "green" product physically handled and evaluated by participants (Study 1), Figure S2: This graphic was used to illustrate the extreme drought in the participants' state, and to connect the meal plan type to environmental conservation (Study 3), Figure S3: The cover story stated that "Green Plan" students would have to display this sticker on their student identification card (Study 3), Figure S4: No interaction was found between environmentalist identity and "green" product preference as a function of social visibility; trend lines shown with continuous 95% confidence intervals (Study 1), Figure S5: No interaction was found between environmentalist identity and "green" product preference as a function of social visibility; trend lines shown with continuous 95% confidence intervals (Study 2), Figure S6: Although identity predicted preference for the Green Plan, there was no interaction between identity and social visibility; trend lines shown with continuous 95% confidence intervals (Study 3).

**Author Contributions:** Conceptualization, C.B.; methodology, C.B. and D.K.S.; software, D.K.S.; formal analysis, C.B. and D.K.S.; resources, D.K.S.; data curation, C.B.; writing—original draft preparation, C.B.; writing—review and editing, C.B. and D.K.S.; visualization, C.B.; supervision, D.K.S.; project administration, C.B.; funding acquisition, D.K.S. All authors have read and agreed to the published version of the manuscript.

**Funding:** This research received no external funding.

**Institutional Review Board Statement:** The study was conducted according to the guidelines of the Declaration of Helsinki, and approved by the Institutional Review Board of the University of California, Santa Barbara, Protocol 31-15-0249 with the most recent modification approved 6 April 2015.

**Informed Consent Statement:** Informed consent was obtained from all subjects involved in the studies, and they remain anonymous.

**Data Availability Statement:** Anonymized data, code, stimuli, and questionnaires are publicly available at https://osf.io/hqnv2.

**Acknowledgments:** We thank the research assistants Ryan Taylor, Jiayi Li, and Taylor Tidwell for data collection, Rachel McDonald for collaboration and data collection, and Heejung Kim for feedback.

**Conflicts of Interest:** The authors declare no conflict of interest. No external funders had a role in the design of the study; in the collection, analyses, or interpretation of data; in the writing of the manuscript; or in the decision to publish the results. This article appears in a special issue edited by the first author, but this article was handled by an editor independent from the special issue.

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
