# Peer review of "When Does Being Watched Change Pro-Environmental Behaviors in the Laboratory?"

_sustainability, doi:10.3390/su13052766_

Round 1
Reviewer 1 Report
Review
*General
This research is experimentally testing a crucial question. Namely the question of whether social identity and related social norms moderate the impact of visibility on pro-environmental behaviors. In three experiments, the authors find a null result. It is crucial that null results are getting published in order to prevent publication bias. Null results, at least when the underlying research design is well-designed, are also advancing our knowledge.
Given a sound research design, there are still several possible interpretations of a null result. Theory is crucial here. What is the exact mechanisms underlying the hypothesis that visibility should moderate identity and related social pressure? A null result speaks differently to different theories and the authors should increase the clarity of their theoretic reasoning (see below) in order to increase the insight provided by their findings.
*Introduction
I think the authors could motivate their research question more strongly. Should social identity and related norms moderate the effect of visibility on pro-environmental behavior – commonly believed to reinforce normative pressure to behave more environmentally friendly – this would mean, that social influence is applicable under less general conditions than usually thought.
[8] is not the only study that found a more favorable perception of environmentally friendly individuals. Another study found that those willing to pay a premium for environmentally friendly products in incentivized buying decisions were trusted more often in incentivized laboratory trust games (Berger, Jo, 2019. Signaling can increase consumers’ willingness to pay for green products, Journal of Consumer Behavior). This is worth mentioning.
The authors point out that identities and norms are both important for pro-environmental behavior. I agree, but I think it is worth making explicit, that both are interrelated. More to the point, different norms could prevail in different social groups, promoting or disfavoring pro-environmental behaviors in those groups. (Kahan, Dan. 2012. “Why We Are Poles Apart on Climate Change.” Nature 488(7411):255; Kahan, Dan M. 2017. “The Expressive Rationality of Inaccurate Perceptions.” Behavioral and Brain Sciences 40).
*Methods
Experiment 2: I suggest to additionally use non-parametric tests and tests for ordinal data (ordered logit). After all, the process of binarization destroys potentially valuable information. Should the results obtained by these tests differ from the results reported, this should be discussed. Should the new results be in line with the results reported, it’s enough to add a sentence on them.
Given the interactions are underpowered, could it make sense to conduced an analysis pooling the data of all three experiments (e.g., using a binary dependent variable)? Or are the three experiments too different?
*Theory
As stated before, a stronger link between clear theoretical mechanisms and results provides more analytical depth. The authors should consider the following theoretical approaches.
A first theory, rooted in evolutionary psychology, states the following. Cues of being watched are activating normative pressures in people. More to the point, they are very careful to comply with groups norms prescribing cooperation, when watched. After all, in our ancestors, non-compliance likely had a sever consequence: ostracism and consequent death. Put differently, visibility automatically triggers a psychologiy of cooperation and compliance (Efferson, Vogt, Berger, Fehr 2015, Eye spots do not increase altruism in children, Evolution and Human Behavior). Under this theory, visibility should have the effect expected by the authors in all three experiments (assuming valid treatments and measures).
Another theory, rooted in economics and classic game theory, hold different implications. As long as the interaction between the participants and the individual watching their behavior is anonymous and one-shot, visibility has no effect – simply because there is nothing to gain with costly compliance in anonymous one-shot interactions. To be more precise, imagine, A, a homo economicus, wishes to maximize his or her personal resources (money, time, effort...). When no one is watching, A is unwilling to take any cost for the sake of the environment. The same holds for anonymous one-shot interactions. In anonymous one-shot interactions, being watched simply doesn’t matter, because there is no reputational cost of not complying/no benefit of complying to group norms prescribing cooperation for the sake of the environment. Only when A’s friends, colleagues, family or neighbors are watching (“shadow of the future”), A is willing to take the cost of compliance (Milinski et al. 2006 Stabilizing the Earth’s climate is not a losing game: Supporting evidence from public goods experiments. PNAS and cited literature. Or, more generally: Axelrod, 1984, the evolution of cooperation).
My suggestion is the following. The authors should reflect the implication of their findings for these theoretical approaches. Experiment 1 contradicts the evolutionary approach, but is consistent with the economic approach. The validity of experiment 2 is questionable, according to the authors. So it’s weak evidence against both approaches. Experiment 3 is evidence against both approaches.
Author Response
Dear reviewer 1,
Thanks very much for your previous comments and please find the enclosed pdf below for specific response.
Best regards,
The authors

Reviewer 2 Report
The paper uses three separate experiments to examine the link between observability/visibility of actions and intensity of pro-environmental behavior and, more importantly, if this link is moderated by environmental identity of the decision-maker.
I think this is an important and interesting topic that has a potential to contribute to our understanding of adaptation practices. It could also improve the design of environmental policy and campaigns.
The authors have done a considerable job conducting three related but distinct studies. They clearly position their paper and relate it to the relevant literature (although there is a larger set of studies on the topic than the paper would let the reader believe). What I also appreciate is the fact that their section 6 discusses virtually all significant shortcomings that I identified while reading the paper.
However, I believe, those shortcomings render the paper virtually uninformative for the scientific audience. There are serious concerns with the construct validity and the convenience-sample (students) nature of the study. But, the most important point is the fact that the paper presents null-results by using underpowered statistical tests. I acknowledge that null results are important and that journals nee to publish more studies containing these. However, for these results to be convincing (and sufficiently informative) authors need to demonstrate the absence of hypothesised effects by appropriately powered statistical tests (and perhaps a more robust statistical methodology [e.g. Bayes factor analysis] than what we find in the current paper).
Author Response
Dear reviewer 2,
Thanks very much for your previous comments and please find the enclosed pdf below for specific response.
Best regards,
The authors

Reviewer 3 Report
I read the paper with pleasure and attach great importance to studies that measure actual rather than self-reported behavior; however, the scientific contribution is limited.
First of all, I agree with the authors that a null result can also be considered a result. Nevertheless, in this paper, we can identify several weaknesses that may explain these null results. Besides, several of these weaknesses are pointed out by the authors in the conclusion of the paper. I agree with each of the limitations identified by the authors, but for the same reason, I do not believe that the paper has sufficient qualities to be accepted. Indeed, if the lack of results is related to a weakness in the experimental design, it is not the same as if it is due to a lack of effect. Especially since this lack of effect is not original and corroborates the results obtained by other research also cited in the paper.
For example, the lack of effect in Study 1 may be because the research assistant made this choice acceptable by offering the product to the participant (green or not). Indeed, the normative aspect of behavior is also related to the social value attributed to it. In this case, the research assistant clearly shows the participant that this choice is acceptable. This is reinforced by concealing aspects of the research.
Finally, environmentalist identity is relatively moderate in the sample. Thus, it is not possible to know if participants have an environmentalist identity, only if they score higher than others on this scale.
In Study 2, the experimental manipulation concerns less the visibility of the behavior than its consequences. Here one could even consider that experimental manipulation modifies the cost of the behavior by making these consequences visible or not (donation + visible commitment). In the same way, I agree with the authors that the environmental aspect of donation is probably overwhelmed by its simple prosocial aspect.
Finally, for study 3, the technical error mentioned does not allow us to measure your research's main hypothesis.
Author Response
Dear reviewer 3,
Thanks very much for your previous comments and please find the enclosed pdf below for specific response.
Best regards,
The authors

Round 2
Reviewer 2 Report
Sorry, and I hate being a typical "reviewer 2" here, but "careful consideration"+dismissal and fear of "substantial delay" in revision is not how (good) science is done.